



# Gas-Particle Partitioning of Semivolatile Organic Compounds When Wildfire Smoke Comes to Town

Yutong Liang[1,2]*, Rebecca A. Wernis[1,3], Kasper Kristensen[1†], Nathan M. Kreisberg[4], Philip L. Croteau[5], Scott C. Herndon[5], Arthur W.H. Chan[1,6], Nga L. Ng[2,7,8], Allen H. Goldstein[1,3]*

[1]Department of Environmental Science, Policy, and Management, University of California, Berkeley, Berkeley, CA, 94720, United States
[2]School of Chemical and Biomolecular Engineering, Georgia Institute of Technology, Atlanta, Georgia 30332, United States
[3]Department of Civil and Environmental Engineering, University of California, Berkeley, Berkeley, California 94720, United States
[4]Aerosol Dynamics Inc., Berkeley, California 94710, United States
[5]Aerodyne Research, Inc., 45 Manning Road, Billerica, Massachusetts 01821, United States
[6]Department of Chemical Engineering and Applied Chemistry, University of Toronto, Toronto, Ontario M5S 3E5, Canada
[7]School of Earth and Atmospheric Sciences, Georgia Institute of Technology, Atlanta, Georgia 30332, United States
[8]School of Civil and Environmental Engineering, Georgia Institute of Technology, Atlanta, Georgia 30332, United States
[†]Now at: Department of Biological and Chemical Engineering, Aarhus University, Finlandsgade 12, 8200 Aarhus N, Denmark

*Correspondence to*: Yutong Liang (yutong.liang@berkeley.edu) or Allen H. Goldstein (ahg@berkeley.edu)

**Abstract.** Wildfires have become an increasingly important source of organic gases and particulate matter in the western United States. A large fraction of organic particulate matter emitted in wildfires is semivolatile, and the oxidation of organic gases in smoke can form lower volatility products that then condense on smoke particulates. In this research, we measured the gas- and particle-phase concentrations of semivolatile organic compounds (SVOCs) during the 2017 Northern California wildfires in a downwind urban area, using the Semivolatile Thermal-Desorption Aerosol Gas Chromatography (SV-TAG), and measured SVOCs in a rural site affected by biomass burning using the comprehensive TAG (cTAG) in Idaho in 2018. Commonly used biomass burning markers such as levoglucosan, mannosan and nitrocatechols were found to stay predominantly in the particle phase, even when the ambient OA was relatively low. The phase partitioning of SVOCs is observed to be dependent on their saturation vapor pressure, while the absorptive equilibrium model underpredicts the particle-phase fraction of most of the compounds measured. Wildfire organic aerosol enhanced the condensation of polar compounds into the particle phase but not some nonpolar compounds, such as polycyclic aromatic hydrocarbons.

## 1 Introduction

In the western United States, wildfires have become an increasingly important source of organic gases and particulate matter in the atmosphere (Dennison et al., 2014; McClure and Jaffe, 2018; Iglesias et al., 2022). Biomass burning (BB) in wildfires emits thousands of organic compounds with vastly different properties (such as volatility, reactivity) (Jen et al., 2019; Hatch et al., 2015; Liang et al., 2022b). Many of these compounds have known direct health impacts, and the oxidation of organic



compounds can also produce secondary organic aerosol (SOA), which are also hazardous (Kim et al., 2018; Tuet et al., 2017;
Wong et al., 2019; Tuet et al., 2019).

Among the wildfire emissions, the semivolatile and intermediate-volatility organic compounds (SVOCs and IVOCs), with
effective saturation concentrations ($C^*$) between $10^{-1}$ to $10^6$ µg m$^{-3}$ are of particular interest because of their partitioning
between the gas and particle phases, which can either decrease or increase the mass concentration of organic aerosol (OA,
which only refers to the particles hereafter) (Hodshire et al., 2019). May et al. (2013) found that the majority of primary organic
aerosol (POA) emitted from BB is semivolatile. In source apportionment analysis, an ideal tracer for a source-specific factor
should be a stable compound with a relatively low volatility compared with the bulk BBOA (Donahue et al., 2012). However,
levoglucosan, the most commonly used BB marker, was also found to evaporate and photochemically decay in the atmosphere
(Hennigan et al., 2010; Xie et al., 2014). Through in-situ measurement, Palm et al. (2020) found that up to a third of the BB
POA evaporated as a wildfire smoke plume diluted by a factor of 5-10, and the majority of SOA formed was from the oxidation
of evaporated BB POA. However, the molecular identities of these evaporated compounds are not known. The BB SOA
compounds formed by atmospheric oxidation of primary emissions may also be semivolatile. For instance, Palm et al. (2020)
also found that the nitrophenolic compounds, which are often assumed to be in the particle-phase only (Finewax et al., 2018),
may also exist in the gas phase. These knowledge gaps motivated our study of the gas-particle partitioning behaviors of BB
POA and SOA compounds in the atmosphere.

When wildfire smoke enters the urban area, the gases and particles in the smoke plumes interact with urban air pollutants. In
absorptive partitioning theory that is assumed in most atmospheric models, the fraction of organic compound in the particle
phase increases with the mass of absorbing aerosol (Pankow, 1994; Donahue et al., 2006). Therefore, the massive amount of
BBOA (which can exceed 100 µg m$^{-3}$ of fine particulate matter) entering the urban atmosphere is expected to absorb SVOCs
into the condensed phase to form SOA. The BBOA may also affect the downwind gas-particle partitioning of compounds such
as polycyclic aromatic hydrocarbons (PAHs) and phthalates in the urban area, altering the mechanisms and magnitude of their
deposition in the respiratory tract (Pankow, 2001). However, whether preexisting organic aerosol can alter the partitioning
behavior of SVOCs also depends on the aerosol composition (Asa-Awuku et al., 2009; Ye et al., 2016, 2018; Liu et al., 2021).
Differences in composition between urban aerosol and BBOA may make them (or some components of them) not miscible.
Mahrt et al. (2022) recently reported that two types of organic aerosol with O/C difference under 0.265 can probably mix with
each other, while two phases will remain if the difference of O/C is above this threshold. BBOA can have O/C ratios above
0.6, while cooking OA and hydrocarbon-like OA, which accounts for nearly half of the urban OA in Oakland, CA (a city
adjacent to Berkeley) have O/C ratios between 0.01 and 0.11 (Zhou et al., 2017; Shah et al., 2018). It is possible that the polar
BBOA compounds can increase the activity coefficients of the relatively nonpolar organic compounds and expel them into the
gas phase. In addition, BBOA was found to be semisolid, even under high relative humidity (RH, ~75%), which can limit the



gas-particle partitioning of organic compounds (Bateman et al., 2017). Measurements are therefore needed to find out whether wildfire aerosol can shift the gas-particle partitioning of SVOCs.

In this study, we measured the hourly concentrations of organic compounds using the SemiVolatile Thermal-desorption Aerosol Gas chromatography (SV-TAG) in Berkeley, California during the October 2017 Northern California wildfires. Concentrations of organic compounds were also measured in McCall, Idaho in August 2018, as a part of the Fire Influence on Regional to Global Environments and Air Quality (FIREX-AQ) study, using the comprehensive Thermal-desorption Aerosol Gas chromatography (cTAG). The TAG instruments can achieve molecular speciation at hourly time-resolution, which enables

us to better explore the impact of environmental factors on the gas-particle partitioning of individual compounds. Our main aims here are to examine the gas-particle partitioning behaviors of organic compounds, especially BBOA marker compounds, when BB smoke is diluted and enters the urban atmosphere, and to evaluate the impact of the massive amount of wildfire aerosol on the partitioning behaviors of SVOCs (including IVOCs, hereafter).

## 2 Materials and Methods

### 2.1 Field sites, measurement periods and the wildfires

The 2017 Northern California wildfires started on October 8 in Napa and Sonoma Counties and lasted for more than 8 weeks. The fuels and perimeters of these wildfires can be found in our previous publications (Liang et al., 2021, 2022a). These wildfires caused multiple Air Quality Measurement Stations in the San Francisco Bay Area to record $PM_{2.5}$ (particulate matter with diameter less than 2.5 µm) exceeding 100 µg m$^{-3}$ (Liang et al., 2021). These air masses contained fire emissions from

mainly oak woodland landscapes combined with suburban housing developments, and agricultural areas. Our SV-TAG measurements were conducted on the UC Berkeley campus (37°52'24.5" N, -122°15'48.2" W), located in the urban San Francisco Bay Area, 55-65 km downwind of the major wildfires. SV-TAG data are reported for October 10 to October 19.

Data were also collected by the cTAG in rural Idaho between August 12 to 27, 2018. The McCall, Idaho field site (44°52'18.5"

N 116°06'55.7" W) was a rural location approximately 4 km south of the town of McCall (population < 3,500) located 180 km north of Boise, surrounded by national forest. This site experienced smoke from 4 fires: Rattlesnake Creek (~3,300 ha, ~45 km northwest of McCall), Mesa (~14,000 ha, 38 km southwest of McCall) and Rabbit Foot (~15,000 ha, ~145 km east of McCall), as well as a fire 60 km southwest of McCall and fires from central Oregon. More detailed descriptions of these fires can be found in Lindsay et al. (2022). Backward trajectory analysis by Lindsay et al. suggests the smoke plumes travelled at

least 3-5 hours and up to 18-30 hours prior to arriving at this site.





## 2.2 Speciated Measurements by SV-TAG and cTAG

During the 2017 California wildfires, individual compounds were measured by the SV-TAG with in-situ derivatization at hourly time resolution (Zhao et al., 2013a; Isaacman et al., 2014). Ambient air was pulled to the SV-TAG at a flow rate > 1000
Lpm through a 2 m × 6 inch (15.2 cm) outer diameter duct to reduce condensation of gas-phase SVOCs in the tubing. The SV-TAG subsampled 20 Lpm of air from the center of the air stream through passivated tubing and a 1.0 µm cutoff (PM$_1$) cyclone. Then 10 Lpm of air was delivered to each of the two identical coated metal mesh filter cells, with one of the air samples passing through a carbon denuder that removes gases. This approach enables us to determine the particle-phase fraction ($F_p$) of each measured compound.


The same strategy was used for cTAG sampling in McCall, Idaho. The cTAG subsampled 10 Lpm of air from the middle of a 25 cm diameter duct, through a PM$_1$ cyclone. The details of the cTAG can be found in Wernis et al. (2021). The cTAG measured VOCs and SVOCs concurrently, and this research focuses on the SVOCs. The main difference between the cTAG and the SV-TAG is that the cTAG alternated between gas + particle and particle-only sampling by either having the sampled
air passing through the denuder (same as the denuder for the SV-TAG) or bypassing the denuder. Since gas + particle and particle-only organics were not collected simultaneously, we calculated the $F_p$ (two-hour time resolution) of each compound using the interpolation method:

$$F_{P,n} = \frac{C_{P,n}}{(C_{G+P,n-1} + C_{G+P,n+1})/2} \tag{1}$$

where $C_{P,n}$ is the particle phase concentration (proportional to the internal-standard-normalized signal) for sample $n$, the
denominator is the sum of gas + particle concentrations in the previous hour and the subsequent hour (Zhao et al., 2013b). More materials and methods about SV-TAG and cTAG can be found in the Supplement.

## 2.3 Uncertainty of Measurements

The uncertainty of SV-TAG measurements has been studied in Isaacman et al. (2014). Uncertainty in quantification of compounds was estimated to be 20-25%. However, because $F_p$ is the ratio of the signals from the two cells, the uncertainty of
$F_p$ only depends on how well samples from the two cells can be intercompared. We collected 11 bypass samples simultaneously on two cells as a direct comparison during this campaign to equalize the signal of the two cells. This equalization helps to reduce the cell-to-cell variabilities caused by differences in collection efficiency, derivatization efficiency, etc. The uncertainty of $F_p$ is estimated to be 15-25% when this correction is applied, so a value of $F_\mathrm{p}$ greater than 1 is possible (Isaacman et al., 2014). For a particle-phase dominant compound, we expect its $F_\mathrm{p}$ to follow normal distribution with a mean value close to 1
(Isaacman et al., 2014). Another source of uncertainty comes from carryover from previous sample, which is found to be 2.6 % by total ion chromatogram and 2.4% by median single ion chromatogram, independent of volatility (Kreisberg et al., 2014). We evaluated the bias caused by carryover in the Supplement. This level of carryover can create a noteworthy positive bias to



the $F_p$ of compounds almost entirely in the gas phase ($F_p$ less than 0.05). Those compounds are not considered in the activity coefficient analysis. The measurement uncertainty of the campaign average $F_p$ can be estimated from the error of the individual

$F_p$ by the central limit theorem, which yields 1.4-2.4% of uncertainty to the campaign average $F_p$ (in Figure 1).

## 2.4 Supporting Measurements and Models

During the 2017 Northern California wildfires, PM$_{2.5}$ filter samples ($N = 74$) with 3 to 4-hour time resolution were collected concurrently with SV-TAG measurements. The chemical composition of the filter samples were determined using two-dimensional gas chromatography coupled with high resolution time-of-flight mass spectrometry (GC×GC HR-ToF-MS)

(Liang et al., 2021). Compounds identified from SV-TAG measurements were validated by matching with authentic standards and comparison with GC×GC measurements. More details can be found in the Supplement. In addition, particle mass concentration was measured by a Scanning Mobility Particle Sizer (TSI 1080 electrostatic classifier coupled with TSI 3788 condensation particle counter, range: 10 – 600 nm, sampling time = 300 s, aerosol flow rate = 0.6 Lpm, sheath flow rate = 6 Lpm), assuming density of PM$_1$ = 1.1 µg m$^{-3}$ according to Pokhrel et al. (2021), and averaged to the same sampling time base

as SV-TAG measurements. The mass of PM$_1$ was also measured by a Grimm 11-A OPC (GRIMM Aerosol Technik) for a shorter period. The linear fit of SMPS measurement against GRIMM measurement gives a slope of 0.98 and normalized root mean square error of 26%, which suggests the density assumed is reasonable. The mass of OA was not measured in Berkeley. In biomass burning emission, OA was found to contribute 77%-99% to total aerosol mass (Lim et al., 2019). We therefore assume that 90% of PM$_1$ is organic in the following analysis. This assumption introduces some uncertainty but did not affect

the conclusion. More discussion about the influence of this assumption is in the Results and Discussion section.

In the FIREX-AQ 2018 campaign in McCall, the bulk properties of non-refractory PM$_{2.5}$ were determined by an Aerosol Chemical Speciation Monitor (ACSM) using thermal vaporization and electron impact ionization mass spectrometry (Ng et al., 2011). The ACSM data were also averaged to the cTAG measurement sampling times. ACSM data and meteorological

data in McCall can be accessed from Yacovitch et al. (2022).

Observed $F_p$'s were compared with the values predicted based on the organic aerosol mass and vapor pressure of each compound, by an absorption equilibrium partitioning model which assumes instantaneous equilibrium between the gas and particle phases (Pankow, 1994). With this assumption, $F_p$ of compound $i$ can be expressed as:

$$F_{p,i} = (1 + \frac{\gamma_i C_i^o}{C_{OA}})^{-1} = (1 + \frac{C_i^*}{C_{OA}})^{-1}$$


(2)

where $C_{OA}$ is the concentration of organic aerosol, $\gamma_i$ is the unitless activity coefficient of compound $i$, $C_i^o$ is the saturation mass concentration of pure compound $i$, and $C_i^*$ is the effective saturation concentration of $i$ (all concentrations are in µg m$^{-3}$). The activity coefficient captures the non-ideal interactions between the compound with the aerosol mixture. A $\gamma$ smaller



than 1 means compound $i$ is more likely to condense into the aerosol mixture than into pure compound $i$. A larger $\gamma$ means the

compound is more likely to stay in the gas phase over this aerosol mixture, and very large $\gamma$ may indicate phase separation (Liu

et al., 2021). The subcooled liquid vapor pressure of each compound, which is used to calculate $C_i^o$, were calculated by the

SIMPOL model and the EVAPORATION model (Pankow and Asher, 2008; Compernolle et al., 2011). To better understand

the influence of BBOA on the gas-particle partitioning of SVOCs, we modeled the activity coefficients of compounds above

aerosol mixtures using the AIOMFAC model (Zuend et al., 2011). More details of these models are provided in the

Supplement.

## 3 Results and Discussion

### 3.1 Average partitioning behavior of BBOA marker compounds

The campaign average $F_p$ for each compound measured during the 2017 Northern California fires are shown in Figures 1 and

S3 (alkanes and aliphatic acids), with distribution of $F_p$ shown in S4. The overall trend is as expected: the gas-particle

partitioning behavior is related to the volatility of compounds. In contrast to Xie et al. (2014), primary BB tracers levoglucosan

and mannosan were found to be almost entirely in the particle phase. The same result is observed in the McCall data (Figure

S6). The discrepancy between our results and Xie et al. is likely due to the much higher mass of BBOA and levoglucosan, and

the lower temperature in our field campaigns. In Xie et al., gas-phase levoglucosan is comparable to particle-phase

levoglucosan only in samples with less than 50 ng m$^{-3}$ of levoglucosan at high ambient temperatures. These primary BB

compounds have much higher $F_p$ compared with other compounds with similar saturation vapor pressures. That makes them

very good BB marker compounds for source apportionment studies. Surprisingly, dehydroabietic acid, although having a very

low saturation vapor pressure over pure compound, has a higher fraction in the gas phase compared with levoglucosan and

mannosan in FIREX-AQ 2018 samples (Figure S5). Nevertheless, it was predominantly in the particle phase in the Napa study.

Retene (C$_{18}$H$_{18}$) is one of the most abundant PAHs emitted from BB (Jen et al., 2019; Liang et al., 2022b). In fresh BB emission

where the level of OA is several hundreds of µg m$^{-3}$, retene is expected to be mainly in particle phase. In both campaigns, the

average $F_p$ of retene was around 0.4-0.5, which suggests a substantial amount of retene was in the gas phase in the aged smoke

plumes. This agrees with our results in the FIREX-AQ 2019 study, in which we found that the ratio of particle-phase

PAHs/Organic Carbon decreased with increasing acetonitrile/furan ratio (with correlation coefficient $r = -0.60$), an indicator

of plume age (Liang et al., 2022b). The particle-phase PAHs, dominated by the diterpenoid-related ones such as retene, on

average accounted for 7.6% of quantified OA in the FIREX-AQ 2019 filter samples but only accounted for less than 0.3% in

the Berkeley filter samples collected simultaneously with the SV-TAG measurements (Liang et al., 2021, 2022b). Evaporation,

in addition to the difference in biomass fuels, may help to explain this observed difference between fresh and aged BBOA.

Given its low O/C ratio and high molecular weight, the evaporated retene (and other diterpenoid-related PAHs) may react with

oxidants and form SOA with a high yield (Gentner et al., 2012; Jimenez et al., 2009). Figure 1 also shows that isomers with



very similar structures and therefore similar estimated saturation concentrations may have very different gas-particle partitioning behaviors. Pyrogallol (1,2,3-benzenetriol) has much higher fractions in the particle phase than 1,2,4-benzenetriol. It is either because the actual vapor pressures of these isomers are very different, or pyrogallol has specific interactions with the aerosol phase that their isomers do not have. Such interactions can reduce the activity coefficients of these SVOCs. Catechol also shows a much higher $F_p$ than hydroquinone. However, catechol could come from the decomposition of particle-phase

dominant compounds in GC. For example, protocatechuic acid (without catalyst, at 200 °C) is shown to produce catechol when heated in water (in CAS SciFinder, https://scifinder.cas.org/). Further studies on the thermal stability of aromatic compounds in thermal desorption systems are needed to elucidate the partitioning behavior of such compounds.

       Secondary BBOA markers, 4-nitrocatechol, 3-methylnitrocatechol and 4-methylnitrocatechol were predominantly in the

particle phase. The sum of their concentrations reached 1.4 µg m$^{-3}$ in the particle phase in the strongest wildfire plumes. The high $F_p$ and high concentration make them good BB SOA markers. Absorptive equilibrium modelling using the vapor pressure estimated by US EPA's EPI Suite (US EPA, 2012) also predicted these compounds stay mostly in the particle phase, while modelling using vapor pressure from the SIMPOL model predicted half of these compounds would stay in the gas phase. The high $F_p$ of these compounds agrees with the findings by Fredrickson et al. in a recent chamber study, in which they found the

group contribution methods overestimate the saturation vapor pressure of these compounds (Fredrickson et al., 2022). Fredrickson et al. also reported that NO$_3$-derived nitrocatechols are less volatile than the OH-derived nitrocatechols (C* = 2.4 µg m$^{-3}$ vs 12 µg m$^{-3}$) at 295K, potentially due to the difference in the composition of the bulk SOA. However, there is no obvious diurnal trend in $F_p$ of nitrocatechols in our study.







**Figure 1**. Campaign average $F_p$ during the October 2017 Northern California wildfires measured by the SV-TAG, classified by functional groups, plotted against their logarithmic saturation concentrations $C^O$ (in µg m$^{-3}$) at 298K. Only a subset of the measured compounds are labeled. Compounds discussed in the text are in bold. Alkanes and mono-carboxylic acids are shown separately in Figure S3.

To test how well the equilibrium absorptive partitioning model explains the partitioning behavior of individual compounds, we calculated the median C* of each compound using Equation 2, converted them to 298K values (see details about the conversion in the Supplement), and plotted it against the estimated $C^O$ over pure compound (Figure 2). Only compounds with $C^O$ between $10^{-2}$ and $10^4$ µg m$^{-3}$ were considered here, because the partitioning method may not work for very volatile or nonvolatile compounds (Stark et al., 2017). It is worth noting that the results in Figure 2 are not very sensitive to the assumed fraction of OA in PM$_1$, because C* is proportional to $C_{OA}$. A 10% change in $C_{OA}$ will therefore only change $\log_{10}(C^*)$ by ~0.04. As has been shown in our previous work, in October 2017, the level of organic carbon (OC) increased by ~6 times during the





wildfires, and the temporal variation of levoglucosan correlated well with OC, and thus BBOA (Liang et al., 2021). We therefore classified the data points into 5 concentration bins based on the levels of particle-phase levoglucosan concentrations, which is a proxy of BBOA influence (Table 1). Under both strong BBOA (high to extreme BBOA in Table 1) and weak BBOA

(low to light BBOA in Table 1) scenarios, the majority of the data points fall between the lines of $\gamma = 1$ and $\gamma = 10^{-3}$. The result suggests these compounds have higher $F_p$ than what is predicted by the models. A few exceptions include diethylhexyl phthalate (DEHP), dibutyl phthalate, diisobutyl phthalate and homosalate, which were found to be more volatile than expected. The low $F_p$ of DEHP is likely related to DEHP in coarse particles (Ma et al., 2014), which is not measured in this study. However, this reason cannot fully explain the order of magnitudes deviation of measured C* from prediction. Many oxygenated

compounds with at least two -OH groups were found to be BB SOA in these fire plumes in our previous study (Liang et al., 2021). In contrast to the GoAmazon wet season study in which they found the $F_p$ of these compounds do not correlate with the vapor pressure (Isaacman-VanWertz et al., 2016), we saw $F_p$ and C* of these compounds qualitatively follows the absorptive equilibrium partitioning models (Figures 1 and 2), though the dependence on $C^O$ is not as strong as alkanes and carboxylic acids (Figure S3). This is likely related to high concentration of organic aerosol (mainly from biomass burning, average

estimated OA = 20.1 µg m⁻³), the dominance of organic aerosol in biomass burning emissions (Lim et al., 2019; Fine et al., 2004), and relatively low average RH (47%) in this study compared with the GoAmazon study (wet season average OA = 1.2 µg m⁻³ with various sources, RH = 90%) (De Sá et al., 2018; Isaacman-VanWertz et al., 2016). SOA formation catalyzed by inorganic compounds such as sulfate and water, which was observed in the GoAmazon study (Yee et al., 2020), was probably less important during the wildfire events in this study.


**Table 1.** Mean and standard deviation of concentrations of levoglucosan, PM₁ and levels of temperature and RH during different levels of BBOA influence in Berkeley during the 2017 Northern California wildfires study.

| Level | Percentiles of particle-phase levoglucosan | Mean ± SD levoglucosan [µg m⁻³] | Mean ± SD, PM₁ [µg m⁻³] | Mean ± SD temperature ºC | Mean ± SD RH |
|---|---|---|---|---|---|
| Low BB | 0-40% | 0.03 ± 0.02 | 4.7 ± 2.4 | 17.5 ± 3.8 | 49% ± 22% |
| Light BB | 40%-60% | 0.18 ± 0.07 | 12.0 ± 4.6 | 15.8 ± 2.9 | 44% ± 20% |
| Medium BB | 60%-75% | 0.72 ± 0.21 | 24.9 ± 9.0 | 16.0 ± 1.5 | 45% ± 24% |
| High BB | 75%-90% | 1.36 ± 0.21 | 36.1 ± 18.3 | 16.3 ± 2.2 | 36% ± 22% |
| Extreme BB | 90%-100% | 2.61 ± 0.43 | 56.3 ± 21.0 | 14.8 ± 1.5 | 45% ± 20% |




Comparing Figures 2A and 2B, we found that the gas-particle partitioning behaviors of compounds are better described by the

model in the strong BB case, especially for compounds with $\log_{10}C^O$ between 1 and 3 (data points are closer to the $\gamma = 1$ line, also shown in Figure S7), which is probably due to the dominance of OA in the strong BB case. When this area is not affected by wildfire smoke, inorganic aerosol accounts for a larger fraction of total aerosol mass (Shah et al., 2018). Inorganic compounds, liquid water in aerosol, and black carbon can therefore have stronger effects on the partitioning of the SVOCs in the low BB case. In addition, although $\log_{10}C^*$ has a roughly linear relationship with $\log_{10}C^O$, the slope is only 0.35 even under

the strong BB scenario. This nonideal behavior is in agreement with the trend observed in multicomponent temperature-programmed desorption experiments, in which researchers found that larger dicarboxylic acids have higher $\gamma$ than smaller ones (Cappa et al., 2008). Together, these results suggest that the inclusion of activity coefficients is needed to better predict the partitioning behaviors of individual organic compounds.

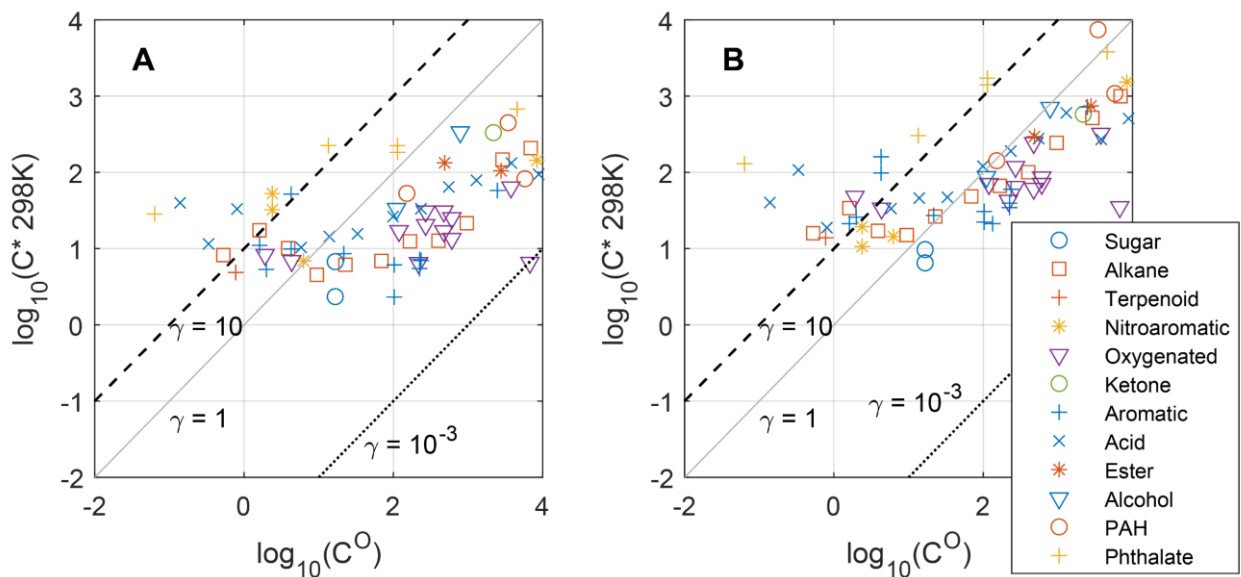

**Figure 2.** The median effective saturation concentration C* (converted to 298K) of each compound measured during the October 2017 Northern California wildfires vs. the estimated saturation concentration over pure compound $C^O$ at 298K, on a logarithmic scale. Lines show the theoretical C* as a function of $C^O$. **A.** Weak BBOA scenario **B.** Strong BBOA scenario.

**3.2 Temporal variations of $F_p$ and factors influencing $F_p$ of individual compounds**

To explore the temporal variations of individual compounds, we first plotted the correlation matrix of the $F_p$ time series. The $F_p$ of lighter alkanes (C13-C20) correlates well with each other, while semivolatile alkanes (C22-C26) form a separate group (Figure S8). This is probably because the lighter alkanes are so volatile that they do not respond to the levels of ambient aerosol like the semivolatile compounds. As shown in Figure 3, the $F_p$ of most PAHs have an anticorrelation with the $F_p$ of oxygenated



compounds. Similar trends are observed in the FIREX-AQ 2018 study (Figure S9), in which the partitioning behaviors of
oxygenated compounds were not well-correlated with compounds without -OH groups. This is likely related to the varying
ambient aerosol composition. Whether preexisting organic aerosol can alter the partitioning behavior of SVOCs depends on
the aerosol composition, because different components of organic aerosol are not always miscible (Ye et al., 2016; Mahrt et
al., 2022b, a). Relatively nonpolar compounds in the ambient aerosol may change the OA matrix, enhancing the absorption of
nonpolar into the particle phase, while they hardly enhance the condensation of polar compounds. In addition, the $F_p$ of methyl
palmitate and methyl stearate correlated very well with each other, but their $F_p$ correlated poorly with the $F_p$ of other
compounds (Figure 3). Methyl palmitate and methyl stearate are emitted from cooking (Kristensen et al., 2019). Their $F_p$ were
elevated (making their γ closer to 1) in the evening, and during breakfast or lunch hours (Figure S10), which suggests the
cooking aerosol enhanced the absorption of fatty acid esters into the particle phase.

**Figure 3.** Correlation matrix of $F_p$ time series for selected compounds observed by SV-TAG in Berkeley, California between
Oct 10, 2017, and Oct 19, 2017. Compounds are ordered by the number of -OH and -COOH groups.




We further explored the importance of environmental factors on the $F_p$ of each compound using a random forest model implemented in MATLAB R2022a. Details of the model can be found in the Supplement. Random forest algorithm is an ensemble approach that makes prediction of the response ($F_p$ in this study) with predictors (measured variables such as temperature) using the aggregation of multiple decision trees. In the 2017 Northern California fires, constrained by the availability of data, we only used time of day, RH, temperature, $PM_1$ and particle-phase levoglucosan concentration as predictors, and $F_p$ of the compounds as the response. Possibly due to the lack of predictors, we were only able to achieve $R^2 >$ 0.45 for predicting the $F_p$ of out-of-bag data (data points not used in individual regression trees) of 4 compounds (fluorene, 4-nitrophenol, 1,2,4-benzenetriol and pentanedioic acid) out of 89 compounds considered. The importance of predictors was estimated by permutation of out-of-bag predictor observations (Figure S11). For fluorene, levoglucosan concentration is the dominant predictor with a negative relationship. For other compounds, $PM_1$ is the most important predictor (with a positive relationship). We also applied this model on $F_p$ measured in the FIREX-AQ 2018 study, in which more predictors are available. Out of 30 compounds considered, 5 of them were predicted with $R^2 > 0.45$. As shown in Figure 4, the level of OA is the dominant predictor of $F_p$ for 2-methylglyceric acid and glyceric acid, while for the two C5 alkene triols, temperature and sulfate concentration become the most important factors. As temperature increases, their $F_p$ decreases ($r = -0.53$ between the $F_p$ of C5 alkene triol-1 and temperature, and $-0.67$ for C5 alkene triol-2). The positive association of sulfate on the $F_p$ of C5 alkene triols, and the positive association of aerosol liquid water content (ALWC) on the $F_p$ of pinonaldehyde and C5 alkene triols are consistent with the catalytic roles of sulfate and water in the formation of these compounds (Yee et al., 2020; Jenkin et al., 2000). Nevertheless, the campaign-average particle-phase OA was 13 µg m$^{-3}$, while the campaign-average ALWC was only 1.5 µg m$^{-3}$, indicating dissolution in aerosol liquid water was not the dominant gas-particle partitioning mechanism for most compounds considered in this study.





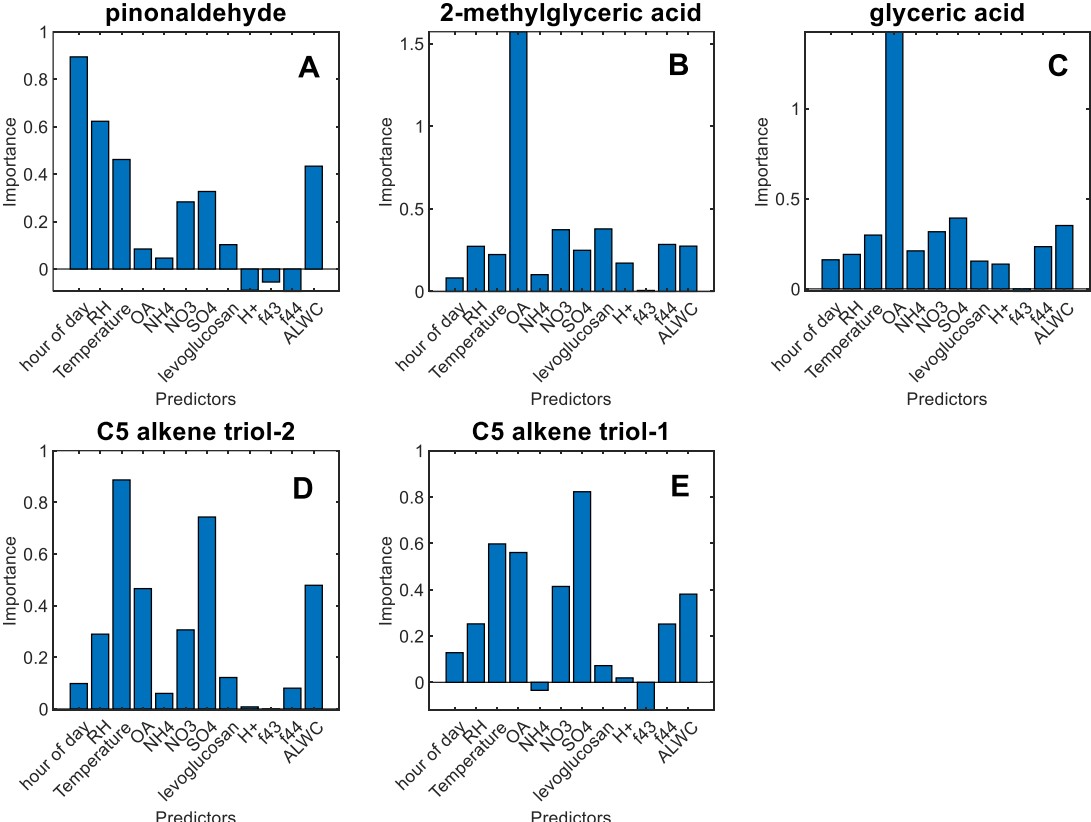

**Figure 4.** The importance scores of factors on the $F_p$ of selected compounds, measured in the FIREX-AQ 2018 study. A high
importance score means the factor is important. Importance scores close to 0 and negative importance scores suggest low
importance. Full descriptions of these predictors can be found in the Supplement.

### 3.3 Effect of BBOA on gas-particle partitioning of SVOCs

When BBOA dominated the ambient aerosol, the gas-particle partitioning of many compounds were different from clean
periods. But such changes are not all in the same direction. Figure 5 shows the $F_p$ of nonpolar compounds decreased with the
level of wildfire smoke influence during the 2017 Northern California wildfires. The $F_p$ of dimethylnapthalene-1 and 2-
methylnaphthalene also decreases with increasing BBOA ($p = 0.021$ and $0.026$, respectively, for the linear fit between mean
BBOA percentile in each bin and mean $F_p$ in each bin), although the changes are smaller. Conversely, levoglucosan, mannosan,
4-nitrocatechol and hexanedioic acid, each with at least two -OH groups, did not follow this trend (Figure S12). Environmental
factors such as temperature and RH remained very similar at different BB influence levels in Berkeley (Table 1). They are
therefore unlikely major causes for this trend. The high O:C ratios of the BBOA and the low O:C ratios of these nonpolar





compounds, which makes them not miscible, is a more plausible cause of the decrease of their $F_p$ during the wildfires. We also compared the $F_p$ of these compounds in different ranges of PM concentrations, and a slightly weaker though still clear dependence of $F_p$ on PM concentration is observed for phenanthrene and fluorene (Figure S13). The slightly weaker

dependence of $F_p$ on PM than on levoglucosan also suggests BBOA is different from local PM, and it caused the decrease of $F_p$ of these nonpolar compounds. Different effects of BBOA on gas-particle partitioning behaviors on relatively polar and relatively nonpolar compounds are also seen in the FIREX-AQ 2018 study. The median $F_p$ of the polar compounds increased almost monotonically from the low BB bin to the high BB bin, while the $F_p$ of the less polar compounds did not (Figure S14). Nevertheless, we did not see the $F_p$ of these nonpolar compounds decrease monotonically when BB influence increased. It is

probably because the aerosol in McCall, Idaho did not have many contributions from urban sources such as traffic and cooking emissions, and the background aerosol (when BB influence was low) had similar compositions with the BBOA that impacted McCall. This is supported by the narrow distribution of $f_{44}$ (the fraction of the OA mass spectrum signal at $m/z$ 44) measured by the ACSM and therefore also the O/C ratio (Aiken et al., 2008) (Figure S15). Thus, the activity coefficients of these nonpolar compounds did not change as much as in the 2017 California study.


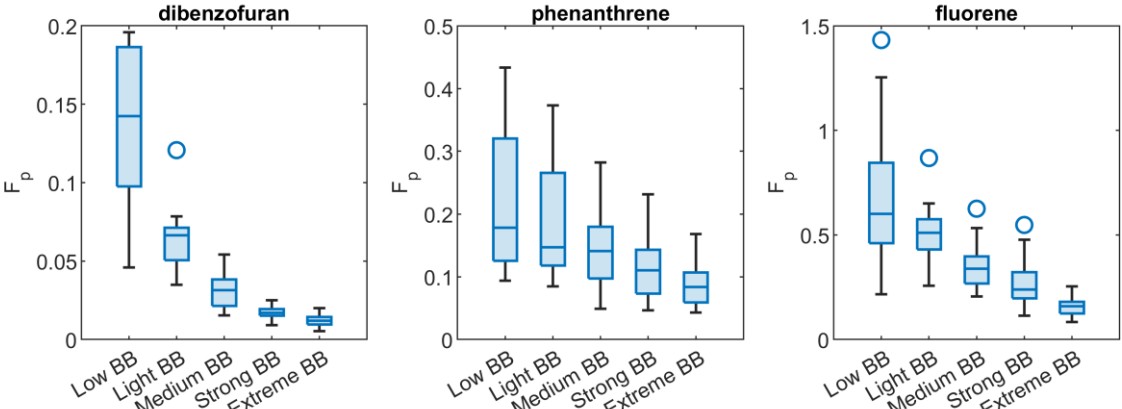

**Figure 5.** Boxplots showing the effect of BBOA on the particle phase fraction on some nonpolar compounds measured in the 2017 Northern California wildfires study. Each box plot shows the interquartile range with whiskers extended to 1.5 × the interquartile range. Central horizontal lines show the medians. Circles denote outliers.


In the 2017 California study, from the low BB scenario to the high BB scenario, the level of $PM_1$ and therefore $C_{OA}$ increases by 10 times ($C_{OA}$ might have increased by more than 10 times because a smaller fraction of local $PM_1$ is OA). According to Equation 2, to make the $F_p$ smaller, $\gamma$ must increase by more than 10 times. We simulated the $\gamma$ of these nonpolar compounds using the AIOMFAC model (in the Supplement). The model successfully predicted that $\gamma$ of these compounds increases as the

fraction of BBOA in total ambient aerosol increases. However, the increase was predicted to be approximately a factor of 2 only (Figure S16). It is worth noting that the simulations were based on assumed composition of aerosol because many





properties of the aerosol (e.g., bulk composition, liquid water content) were not measured in Berkeley, which introduced uncertainty to the simulations.

The high viscosity of BBOA may contribute to the γ of these compounds increasing with fraction of BB influence. Under the RH and temperature in Berkeley during these fires, the BBOA is expected to be semi-solid (Li et al., 2020). This can affect the evaporation and condensation of the SVOCs (Zelenyuk et al., 2012). If this effect is important here, compounds mainly emitted in the particle phase from wildfires would have $F_p$ higher than what is predicted from the absorptive equilibrium model, while compounds later condensed into the particle phase (such as SVOCs mainly emitted in the urban area such has phthalates

and fatty acid esters) would have $F_p$ lower than prediction when BBOA dominates. However, we did not find such a trend, which may suggest the effect of viscosity on $F_p$ is overshadowed by other factors.

   The observed partitioning phenomenon of BBOA has at least two interesting implications. First, the high gas-phase fraction of the nonpolar compounds can reduce their atmospheric lifetimes. For instance, the gas-phase lifetime of fluorene and

phenanthrene are 21 hours and 9 hours under a typical atmospheric environment ([OH] = $10^6$ molecules cm$^{-3}$, [O$_3$] = 50 ppb), respectively (Keyte et al., 2013). However, these compounds have much longer lifetimes in the particle phase, because OA can shield them against heterogeneous oxidation and evaporation (Zelenyuk et al., 2012). The low particle-phase fraction of these compounds in the BB plumes can shorten their lifetimes (May et al., 2012), reducing their long-range transport. Nonetheless, the gas-phase PAHs can react with oxidants to form oxygenated and nitrated PAHs, which may have higher

toxicity than the parent PAHs (Idowu et al., 2019). Also, the result implies that PAHs, which have urban sources, do not necessarily partition more to the condensed phase when the urban area is affected by wildfire smoke. The phase distribution of compounds directly affects where they will deposit in the respiratory tract, what kind of cells will be exposed to these compounds, and consequently the health impacts (Liu et al., 2017). However, the PAHs observed in this research are not as toxic as heavy PAHs such as benzopyrenes. Future works on the gas-particle partitioning behaviors of more toxic PAHs are

still needed.

## 4. Conclusions

In this work, we evaluated the measured phase partitioning behaviors of various BB markers. Levoglucosan, mannosan and nitrocatechols were found to stay mainly in the particle phase, even when the ambient OA was relatively low, which suggests they are good marker compounds for source apportionment. The phase partitioning of SVOCs is observed to be strongly

dependent on the saturation vapor pressure, yet whether BBOA is abundant, the absorptive equilibrium model underpredicts the particle-phase fraction of the majority of the compounds measured. BBOA enhanced the condensation of polar compounds into the particle phase but not some nonpolar compounds, such as PAHs.

Although we were able to qualitatively find out the factors influencing the gas-particle partitioning of compounds, with the
current dataset, even using machine learning algorithms, we were not able to accurately predict the partitioning behavior of
the majority of compounds. Future longer-term measurements of gas-phase and particle-phase SVOCs simultaneously with
measurements of environmental factors and bulk aerosol composition may help to further address this problem.

## 5. Data availability

Data used in this research are available from the corresponding authors upon request.

## 6. Author contribution

Y.L., R.A.W., N.K., and A.H.G. designed research. Y.L., R.A.W., K.K., N.K., P.L.C., and S.C.H. performed research. Y.L.,
R.A.W., A.W.H.C., N.L.N., and A.H.G. analyzed data. Y.L. and A.H.G. wrote the manuscript with suggestions from
coauthors.

## 7. Competing interests

Some authors are members of the editorial board of journal Atmospheric Chemistry and Physics. The peer-review process was
guided by an independent editor, and the authors have also no other competing interests to declare.

## 8. Acknowledgements

The authors acknowledge Connor Daube, Francesca Majluf, Joseph Roscioli, Edward Fortner, Christoph Dyroff, Tim Onasch,
Jordan Krechmer and Tara Yacovitch of Aerodyne Research, Inc. and Andrew Lindsay from Drexel University for the
planning, operation, support and analysis of Aerodyne-operated instruments in the field. The authors also thank David
Lunderberg from UC Berkeley for helpful discussions, and for locating and retrieving the SV-TAG data.

## 9. Financial Support

The UC Berkeley team was sponsored by the National Science Foundation (RAPID grant 1810641) and the National Oceanic
and Atmospheric Administration (grant number: NA16OAR4310107). Aerodyne Research, Inc. measurements were supported
by NOAA grants NA17OAR4310102 and NA16OAR4310104.



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
