# Peer review of "Gas-Particle Partitioning of Semivolatile Organic Compounds When Wildfire Smoke Comes to Town"

_EGUsphere, 2023_

## Author Comment (AC1)

Dear Editor and Reviewers,

Thank you for your comments and suggestions, which have served to improve the clarity and quality of this manuscript. Our point-by-point responses to the comments are in blue, and modifications to the text are in brown.

**Reviewer 1**

Liang et al. measured gas-particle partitioning of 89 compounds through SV-TAG and cTAG either by measuring the quantity of each compound in the gas and particle phase, or the particle phase only and taking the difference to obtain the concentration of the compound in the gas phase. By measuring the gas phase and particle phase, the measured fraction in the particle can be compared to the saturation mass concentration to obtain information about gas-particle partitioning for individual compounds in the ambient atmosphere. The main body of the paper looks at gas-particle partitioning in an urban environment with influence from biomass burning. This paper demonstrates that polar compounds partition into biomass burning organic aerosol more readily than nonpolar compounds, which preferentially evaporate into the gas phase with increasing BBOA.

The measurement technique is well-characterized, and supported with additional measurement techniques to ensure unambiguous identification and quantification of the reported compounds. The findings are unique in that they report direct measurements of gas-particle partitioning in the ambient atmosphere with some mechanistic understanding of the change in activity coefficients in

the presence of different organic aerosol. The paper is well organized with appropriate figures.

This paper should be published after responding to the comments below.

We thank Reviewer 1 for the comments and suggestions, which have served to improve the clarity and quality of this manuscript. Our responses and changes to the text and figures are as follows.

General comments:

The analysis, interpretation, discussion and conclusions in this manuscript do not address any uncertainty within the vapor pressure group contribution methods used to estimate the saturation mass concentration of the identified compounds. The interpretation that the activity coefficient is solely responsible for differences between measured and expected fraction of compound in particle makes sense mathematically, but relies on the assumption that the uncertainty in the group contribution methods used is small relative to the total discrepancy observed. Evaluations of the uncertainty of SIMPOL have been conducted in e.g. Bilde et al., **2015**, *Chem. Rev.*, 4115-4156 and Valorso et al., **2011**, *Atmos. Chem. Phys.*, 6895-6910, and there is evidence that vapor pressures of multifunctional compounds tend to be inaccurate when by calculated SIMPOL and other group contribution methods in e.g. Barley and McFiggans, **2010**, *Atmos. Chem. Phys.*, 749-767, Dang et al., **2019**, *Aerosol Sci. Tech.*, 1040-1055, and O'Meara et al., **2014**, *Phys. Chem. Chem. Phys.*, 19453-19469. The error in the calculated saturation mass concentration could produce deviations from gamma = 1 that are not related to the activity coefficient. A discussion of the uncertainty of the group contribution methods is critical for the interpretation of the data presented in this

manuscript, and must be included. The discussion should include whether the uncertainties in the group contribution methods are random or systematic, and how this can impact the conclusions stated in the manuscript.

We thank the reviewer for this comment. We are not claiming that the vapor pressure calculated from the SIMPOL model is accurate, and we are not using the predicted values as the benchmark to evaluate our measured values. We chose the SIMPOL model (and the EVAPORATION model) mainly because they are commonly used for predicting vapor pressure. The predicted values from SIMPOL or EVAPORATION models are what people typically use to describe the gas-particle partitioning of SVOCs. The difference between the predicted and measured $F_p$ values helps us find out whether these models can predict the gas-particle partitioning of these compounds reasonably well. We believe the measurement values can help future models to improve the ability of $F_p$ prediction.

In addition, we did not solely rely on the SIMPOL model in this analysis. As we wrote in the Supplement, the subcooled liquid vapor pressures were estimated by two group contribution models, i.e., the SIMPOL model and the EVAPORATION model and they showed good agreement ($R^2 = 0.97$ and root mean square error of 0.44 of $\log_{10}P^O$). In addition, we used the MPBPWIN component (modified Grain method) in EPA's EPI Suite (US EPA, 2012) to estimate the vapor pressures of nitro-aromatic compounds, because the SIMPOL model substantially overestimates the vapor pressures of these compounds in general (Bannan et al., 2017; Wania et

al., 2017). These steps help to make our predicted vapor pressure more accurate. We moved this information from the Supplement to the main text now.

We also added discussion about the uncertainty to the main text.

From Line 161 of the revised text:

"The subcooled liquid vapor pressure of each compound, which is used to calculate $C_i^o$, were calculated by the SIMPOL model and the EVAPORATION model (Pankow and Asher, 2008; Compernolle et al., 2011). We used the MPBPWIN component (modified Grain method) in US EPA's EPI Suite (US EPA, 2012) to estimate the vapor pressures of nitro-aromatic compounds, because the SIMPOL model substantially overestimates the vapor pressures of these compounds in general (Bannan et al., 2017; Wania et al., 2017). There is evidence that vapor pressures of multifunctional compounds may not be very accurate when by calculated SIMPOL and other group contribution methods (Barley and McFiggans, 2010; Dang et al., 2019; O'Meara et al., 2014). We used these methods to estimate $C^O$ mainly because they are commonly used to describe the gas-particle partitioning of SVOCs."

From Line 224 of the revised text:

"In many studies, such as Isaacman-VanWertz et al. (2016) and Nie et al. (2022), when predicting the gas-particle partitioning of organic compounds using Equation 2, C* is assumed to be the same as $C^O$ ($\gamma = 1$). To test how well the equilibrium absorptive partitioning model with the $\gamma = 1$ assumption explains the partitioning behavior of individual compounds, we calculated the median

C* of each compound from measured $F_p$ using Equation 2, converted them to 298K values (see details about the conversion in the Supplement), and plotted them against the $C^O$ over pure compound estimated by group contribution models (Figure 2)."

Minor Comments:

Pg. 2 line 48: Finewax et al. did not assume that nitrophenol compounds were only in the particle-phase, but measured the saturation mass concentration of 4-nitrocatechol, C* = 13 ug m^-3, and concluded that it was almost entirely in the particle phase of the particle concentrations used in their laboratory study. This was supported by Fredrickson et al., 2022, ACS Earth & Space Chemistry, which reported measured vapor pressure of 2.4 – 12 ug m^-3. Please revise this, and include the Finewax citation in the discussion of 4-nitrocatechol vapor pressure on pg 7 lines 199-208.

We thank the Reviewer for this suggestion. We looked at Finewax et al. (2018) again, and they said their C* is estimated by thermal desorption. We therefore changed this sentence into:

"For instance, Palm et al. (2020) also found that the nitrophenolic compounds, which were predicted to be in the particle-phase only from thermal desorption analysis (Finewax et al., 2018), may also exist in the gas phase."

We also added the following sentence to Line 199-208 as suggested.

"The high $F_p$ of these compounds agrees with the findings by Finewax et al. (2018) and Frederickson et al. (2022), in which they found the group contribution methods overestimate the saturation vapor pressure of these compounds."

Pg. 6-7 Lines 189-193: Please cite literature e.g. Dang et al., **2019**, *Aerosol Sci. Tech.*, 1040-1055, to demonstrate precedent of different isomers having significantly different vapor pressures.

We thank Reviewer 1 for this suggestion. The following sentence has been added.

"Large variation of vapor pressures among isomers has been reported before (Dang et al., 2019)."

Pg. 9 line 225: Are activity coefficients of 10^-3 expected? Please include literature values of gamma to support this finding.

As shown in Figure 2, only 1 compound has an activity coefficient close to $10^{-3}$. The derived activity coefficients of most compounds fall between 1 and $10^{-2}$. We revised the range mentioned in the text. There is limited information about the inferred activity coefficients of SVOCs in ambient aerosol, but we added some discussion on the activity coefficient by citing literature.

"The range of $\gamma$ inferred from this study is similar with what Cappa et al. (2008) found for dicarboxylic acids in multicomponent mixtures. The $\gamma$ inferred for adipic acid (~0.3) is also within the range of $\gamma$ inferred for adipic acid in aidpic acid–ambient extracts mixtures (Saleh and Khlystov, 2009)."

Pg. 10 Figure 2: It's not unambiguous that Figure 2B demonstrates that the gas-particle partitioning of compounds is better described by the model in the strong BB case, merely that compounds are

closer to the gamma = 1 line. This could suggest that predicted vapor pressure based on group contribution methods fail to accurately determine the vapor pressure of multi-functional compounds, or it could suggest that gas-particle partitioning may not be instantaneous for BBOA at these atmospheric conditions (e.g. RH, temperature). It could demonstrate the uncertainty in the group contribution methods for calculating compound vapor pressure. I believe that the statement on pg 10 lines 252-253 does not necessarily follow without also assuming that the uncertainty in the vapor pressure calculations are very small.

We agree that the group contribution method may not be able to determine the vapor pressure accurately. But with due respect, we believe the comparison between the low BB cases and the high BB cases are still valid, because the bias caused by the group contribution method should be almost the same under the low BB scenario and the high BB scenario. Other reasons such kinetic constraints at those RH and T mentioned by Reviewer 1 are possible, but we did not find statistically significant differences of RH and T under difference BB influence categories (Table 1). We would like to revise this section by adding some discussion on viscosity. The discussion of the uncertainty related to the group contribution method is more relevant to the paragraph above, when we first introduced the results of the equilibrium absorption model. It is added there (see our response to the general comments of Reviewer 1).

"Comparing Figures 2A and 2B, we found that the gas-particle partitioning behaviors of compounds are better described by the model in the strong BB case, especially for compounds with $\log_{10}C^O$ between 1 and 3 (data points are closer to the $\gamma = 1$ line, also shown in Figure S7), which is probably due to the dominance of OA in the strong BB case. It is also worth noting that the deviation from the $\gamma = 1$ line can be partly attributed to the inaccuracy in $C^O$ estimated by the group contribution models. When this area is not affected by wildfire smoke, inorganic aerosol accounts for a larger fraction of total aerosol mass (Shah et al., 2018). Inorganic compounds, liquid water in aerosol, and black carbon can therefore have stronger effects on the partitioning of the SVOCs in the low BB case. Also, the equilibration timescale of SVOC between gas and particle phases increases with decreasing mass of PM (Shiraiwa and Seinfeld, 2012). That can also limit the capability of the equilibrium absorption model in predicting the partitioning of many BB SVOCs under relatively clean scenarios."

Pg. 15 Lines 340-346: Viscosity discussion should include the age of the biomass burning organic aerosol. Based on viscosity measurements of BBOA in the literature, and the size distribution of the particles, can the gas-particle partitioning timescale be calculated? Based on the timescale calculated and the age of the smoke plume, the authors should be able to determine whether gas-particle partitioning has reached equilibrium as it approaches the sampling site.

We thank the reviewers for this suggestion. We estimated the chemical plume age using several reactive biomass burning VOCs, the fire plumes traveled at least 2 hours and up to more than 10

hours before reaching Berkeley (Liang et al., 2022). According to Shiraiwa and Seinfeld (2012), for semivolatile compounds in smoke plumes with tens of micrograms of PM, they should have reached partitioning equilibrium if they are from the fire. We added the following discussion of this point.

"However, the ages of the plumes are at least two hours (Liang et al., 2022), which are long enough for these nonpolar compounds to reach gas-particle partitioning even when the particles are mainly in semi-solid state (Shiraiwa and Seinfeld, 2012). If the effect of viscosity is important here, compounds mainly emitted in the particle phase from wildfires would have $F_p$ higher than what is predicted from the absorptive equilibrium model, while compounds later condensed into the particle phase (such as SVOCs mainly emitted in the urban area such has phthalates and fatty acid esters) would have $F_p$ lower than prediction when BBOA dominates."

Technical Comments:

Figure 2: Legend should be placed elsewhere to not block x-axis label.

Figure 2 is changed as suggested.

[Figure]

Figure S13: These compounds are all non-polar. Revise the caption.

We thank Reviewer 1 for catching this mistake. The caption has been changed as suggested.

**Reviewer 2**

The authors determined the particle-phase fraction (Fp) of the individual semivolatile organic compounds measured by two TAG instruments and looked at the gas-particle partitioning behavior of SVOCs under the influences of biomass burning at two locations in two separate years. The manuscript provides insight into the phase partitioning of SVOCs and factors influencing the behavior in wildfire plumes. I recommend the manuscript for publication after addressing the following comments:

We thank Reviewer 2 for the very constructive comments and suggestions. We improved our analysis and writing based on these comments. Our point-by-point responses and revisions are as follows.

Specific comments:

1. Please explain why out of the many SVOCs identified, the authors selectively discussed a few compounds in the text (bold in Figure 1) in Section 3.1. It is not clear what's special about these compounds or their significance; why discuss these over the others. In addition, the authors claimed in Lines 175-176 and lines 200-201 that the higher observed Fp makes levoglucosan and nitrocatechols etc "very good BB marker compounds". I would provide references that these compounds are exclusively emitted/formed from biomass burning with negligible other sources. Please also explain why high Fp makes them good BB markers.

We thank Reviewer 2 for this suggestion. We pick these compounds mainly because they are commonly used as tracers for biomass burning. This is reflected by the title of Section 3.1, "Average partitioning behavior of BBOA marker compounds". We added the following sentence to the manuscript.

"We focus our discussion on commonly used biomass burning markers."

We added two references showing these compounds are uniquely from biomass burning sources.

"In contrast to Xie et al. (2014), primary BB tracers levoglucosan and mannosan (Eliasl et al., 2001; Simoneit, 2002) were found to be almost entirely in the particle phase."

The reason why these compounds are good biomass burning tracers is because if they mainly stay in the particle phase, they have longer lifetime, which can better indicate the influence of biomass burning to the PM at the receptor site (Donahue et al., 2012). We added this reference.

"The high particle-phase fraction of these compounds makes them less likely to react with atmospheric oxidants and decay, which makes them very good BB marker compounds for source apportionment studies (Donahue et al., 2012)."

2.      The evaluation of the performance of the equilibrium absorptive partitioning model would benefit from additional explanation/description. It is unclear what criteria were used to assess the predictive capabilities of the model and why.

From Reviewer 2's comment 2d, we think the reviewer misunderstood how we defined C* and $C^O$. Sorry for the confusion caused. Our point-by-point responses are as follows.

a.  Line 215-217, comparing C* vs C° does not provide insight into the performance of the equilibrium absorptive partitioning model. I would clarify that calculated C* using both the measured Fp and the predicted Fp, and did the comparison between these two sets of results.

Comparing the C* vs $C^O$ should be equivalent to comparing the measured $F_p$ and the predicted $F_p$ because C* is inferred from the measured $F_p$ while $C^O$ is predicted by the equilibrium absorption model. Our $C^O$ does not account for activity coefficient. We revised the text by emphasizing that:

"In many studies, such as Isaacman-VanWertz et al. (2016) and Nie et al. (2022), when predicting the gas-particle partitioning of organic compounds, C* is assumed to be the same as $C^O$ ($\gamma = 1$). To test how well the equilibrium absorptive partitioning model with the $\gamma = 1$ assumption explains the partitioning behavior of individual compounds, we calculated the median C* of each compound from measured $F_p$ using Equation 2, converted them to 298K values (see details about the conversion in the Supplement), and plotted it against the $C^O$ over pure compound estimated by group contribution models (Figure 2)."

b.  Line 225-226. Why does the measured data falling between the lines of γ=1 and γ=10⁻³ in Figure 2 suggest that the model underpredicts the $F_p$?

The original sentence is misleading. We revised the sentence into:

"The result suggests these compounds have higher $F_p$ than what is predicted by the models (if $\gamma = 1$ is assumed)."

If the actual $\gamma$ is smaller than 1, while the model assumes $\gamma = 1$, that means $\gamma$ is overestimated.

From Equation 2, $F_{p,i} = (1 + \dfrac{\gamma_i C_i^o}{C_{OA}})^{-1}$, if $\gamma$ is overestimated, $F_p$ will be underestimated.

c. Line 244-245. It is hard to follow why the data points being closer to the line of $\gamma=1$ suggests a better-described phase partitioning behavior of compounds.

This is related to the issue pointed out by Reviewer 2 in Comment 2b. We are sorry for the confusion caused. We revised the sentence into:

"The result suggests these compounds have higher $F_p$ than what is predicted by the models (if $\gamma = 1$ is assumed)."

d. Line 252-253. The activity coefficients were included in the calculation of predicted C*. I don't see any cases illustrating that excluding activity coefficients leads to a worse prediction.

 Again, we are very sorry about the confusion caused. In this study, C* is inferred from the measured $F_p$ while $C^O$ is predicted by the equilibrium absorption model. Our $C^O$ does not account for activity coefficient. The changes to the text can be found in our responses to Comments 2a and 2b.

3. Did the authors look at the Fp of SVOCs under non-BB periods? How did that compare to different BB scenarios? The title highlighted "when wildfire smoke comes to Town". However, the manuscript only presented/discussed results under BB influences. It will be useful to add the

non-BB scenario to the figures and table (Figure 5, Figure S12 – 15, Table 1) and discussions in Section 3.3 to set the base case of "background" air in Town to reflect the title.

We thank Reviewer 2 for this suggestion. We want to clarify that the non-BB scenario has been included in the results and discussion. As shown in Table 1, our **low-BB** scenario includes the "background" periods. We separated the data into 5 categories, based on the level of biomass burning influence (Table 1). Then in Figure 2, we discussed the differences of C* of compounds under different BB scenarios. In the figures Reviewer 2 mentioned, the background periods are all included.

4.      The manuscript used data collected during two studies: the 2017 Berkeley wildfire and the 2018 McCall FIREX-AQ. However, the main text mainly presented the 2017 Berkely wildfire, and the discussions were mainly focused on the Berkely results too, except for the Random Forest algorithm results where the 2017 Berkely study had limited data. Can the authors discuss the 2018 results more, and comment on the similarities and differences between the 2017 vs 2018 results and any insights gained from comparing the two studies?

We thank Reviewer 2 for this suggestion. We chose not to directly compare the $F_p$ measured in those two campaigns mostly because the measurement techniques are slightly different. For 2017 measurement on campus, we used SV-TAG which simultaneously measures gas+particle and particle-only signals every hour. However, for the 2018 measurement by the cTAG, at each hour, we either measure gas+particle or particle-only signals, and interpolated the data to calculate $F_p$.

That makes $F_p$ from the two campaigns have different uncertainties, which are thus not directly comparable in many cases. We have already included reasonable comparisons of the two studies in the manuscript.

Minor comments:

1.      How many compounds in total were measured/identified from these two data sets? I would state it at the beginning of the results.

This information was originally provided in the Supplement. We changed the text to mention it at the beginning of the results section. The following sentence has been added.

"After QA and QC (Supplement Section S1), we selected 89 compounds from the 2017 study and 30 compounds from the 2018 study for detailed analysis."

2.      Lines 159 – 161, I would be more specific with the range of γ that indicates different phasing. In (Liu et al., 2021), they found that a phase separated ammonia sulfate-SOA mixture has a γ = 74.

We changed this sentence into:

"…very large $\gamma$ (>10) may indicate phase separation (Liu et al., 2021; Donahue et al., 2011)."

3.      Line 203, please clarify what "half of these compounds would stay in the gas phase" mean. Half number, half mass?

Thanks for catching this issue. We modified this sentence into:

"…while modelling using vapor pressure from the SIMPOL model predicted half of these compounds by mass would stay in the gas phase."

4.      Line 322-323. I disagree with the statement. Figure S15 shows that f44 and O/C values under lower BB influences had much broader distributions than those under stronger BB influences. This on the contrary suggests that the background aerosols when BB influence was low had different compositions from the BBOAs at McCall. Did you check the tracers for cooking and traffic aerosols?

For some data points, there might be influence from other PM sources. However, considering the low BB data points together, there is no statistically significant difference of $F_p$ among different groups. In addition, although the f44 distribution for the low BB influence case has larger variation, still, the 25th and 75th percentiles are 0.14 and 0.18, respectively. Also, our site in McCall is in a remote location (44°52'18.5" N, 116°06'55.7" W). We do not expect strong influence of cooking and traffic aerosol. We did not see high concentrations of markers for these pollution sources. Hydrocarbon-like OA (HOA) and cooking OA (COA) usually have f44 < 5% (Mohr et al., 2009; Ng et al., 2010), which is very different from the values we observed. That further indicates relatively small influence of HOA and COA.

We clarify our interpretation as follows.

"This is supported by the narrow distribution of $f_{44}$ (the fraction of the OA mass spectrum signal at $m/z$ 44) measured by the ACSM and therefore also the O/C ratio (Aiken et al., 2008), and the lack of statistically significant difference of $f_{44}$ under difference BB influence levels (Figure S15)."

Technical comments:

1.      Figure 2, I would clarify in the caption that the markers are the C* derived from the measured Fp (C* was not directly measured as stated in line 255), and that the lines are the C* derived from the predicted Fp.

We changed the caption as suggested. Now it reads:

"The median effective saturation concentration C* (converted to 298K) of each compound derived from Equation 2 during the October 2017 Northern California wildfires…"

2.      Figure 3, the upper right half of the matrix is the repetition of the lower left half of the matrix. This is redundant and confusing. I would show only half of the matrix. The same applies to Figure S8

We agree with Reviewer 2 that the upper right parts of the figures are redundant. These figures are changed as suggested.

Figure 3:

[Figure]

Figure S8:

[Figure]

[Figure]

3.    Figure S3, the left axis label should be Fp. The legend of the figure should be modified to

include "Measured" for the markers of Alkane and Acid, and "Predicted" for the lines with

different gamma values.

We changed the left axis label to $F_p$, and changed the caption as follows.

[Figure]

"**Figure S3.** Measured and predicted $F_p$ for *n*-alkanes and *n*-carboxylic acids, against the $\log_{10}C^O$ at 298K in the 2017 Northern California wildfires study. Scatters show the measured $F_p$ and lines show the predicted $F_p$ from the equilibrium absorption model."

**References:**

Dang, C., Bannan, T., Shelley, P., Priestley, M., Worrall, S. D., Waters, J., Coe, H., Percival, C. J., and Topping, D.: The effect of structure and isomerism on the vapor pressures of organic molecules and its potential atmospheric relevance, Aerosol Sci. Technol., 53, 1040–1055, https://doi.org/10.1080/02786826.2019.1628177, 2019.

Donahue, N. M., Epstein, S. A., Pandis, S. N., and Robinson, A. L.: A two-dimensional volatility basis set: 1. organic-aerosol mixing thermodynamics, Atmos. Chem. Phys., 11, 3303–3318, https://doi.org/10.5194/acp-11-3303-2011, 2011.

Donahue, N. M., Kroll, J. H., Pandis, S. N., and Robinson, A. L.: A two-dimensional volatility basis set – Part 2: Diagnostics of organic-aerosol evolution, Atmos. Chem. Phys., 12, 615–634, https://doi.org/10.5194/acp-12-615-2012, 2012.

Eliasl, V. O., Simoneit, B. R. T., Cordeiro, R. C., and Turcq, B.: Evaluating levoglucosan as an indicator of biomass burning in Carajás, Amazônia: A comparison to the charcoal record, Geochim. Cosmochim. Acta, 65, 267–272, https://doi.org/10.1016/S0016-7037(00)00522-6, 2001.

Finewax, Z., De Gouw, J. A., and Ziemann, P. J.: Identification and Quantification of 4-Nitrocatechol Formed from OH and NO3 Radical-Initiated Reactions of Catechol in Air in the Presence of NOx: Implications for Secondary Organic Aerosol Formation from Biomass Burning, Environ. Sci. Technol., 52, 1981–1989, https://doi.org/10.1021/acs.est.7b05864, 2018.

Frederickson, L. B., Sidaraviciute, R., Schmidt, J. A., Hertel, O., and Johnson, M. S.: Are dense networks of low-cost nodes really useful for monitoring air pollution? A case study in Staffordshire, Atmos. Chem. Phys, 22, 13949–13965, https://doi.org/10.5194/acp-22-13949-2022, 2022.

Liang, Y., Weber, R. J., Misztal, P. K., Jen, C. N., and Goldstein, A. H.: Aging of Volatile

Organic Compounds in October 2017 Northern California Wildfire Plumes, Environ. Sci.

Technol., 56, 1557–1567, https://doi.org/10.1021/acs.est.1c05684, 2022.

Liu, X., Day, D. A., Krechmer, J. E., Ziemann, P. J., and Jimenez, J. L.: Determining Activity

Coefficients of SOA from Isothermal Evaporation in a Laboratory Chamber, Environ. Sci.

Technol. Lett., 8, 212–217, https://doi.org/10.1021/acs.estlett.0c00888, 2021.

Mohr, C., Huffman, J. A., Cubison, M. J., Aiken, A. C., Docherty, K. S., Kimmel, J. R., Ulbrich,

I. M., Hannigan, M., and Jimenez, J. L.: Characterization of primary organic aerosol emissions

from meat cooking, trash burning, and motor vehicles with high-resolution aerosol mass

spectrometry and comparison with ambient and chamber observations, Environ. Sci. Technol.,

43, 2443–2449, https://doi.org/10.1021/es8011518, 2009.

Ng, N. L., Canagaratna, M. R., Zhang, Q., Jimenez, J. L., Tian, J., Ulbrich, I. M., Kroll, J. H.,

Docherty, K. S., Chhabra, P. S., Bahreini, R., Murphy, S. M., Seinfeld, J. H., Hildebrandt, L.,

Donahue, N. M., Decarlo, P. F., Lanz, V. A., Prévôt, A. S. H., Dinar, E., Rudich, Y., and

Worsnop, D. R.: Organic aerosol components observed in Northern Hemispheric datasets from

Aerosol Mass Spectrometry, Atmos. Chem. Phys., 10, 4625–4641, https://doi.org/10.5194/acp-

10-4625-2010, 2010.

Palm, B. B., Peng, Q., Fredrickson, C. D., Lee, B. H., Garofalo, L. A., Pothier, M. A.,

Kreidenweis, S. M., Farmer, D. K., Pokhrel, R. P., Shen, Y., Murphy, S. M., Permar, W., Hu, L.,

Campos, T. L., Hall, S. R., Ullmann, K., Zhang, X., Flocke, F., Fischer, E. V., and Thornton, J.

A.: Quantification of organic aerosol and brown carbon evolution in fresh wildfire plumes, Proc.

Natl. Acad. Sci., 117, 29469–29477, https://doi.org/10.1073/pnas.2012218117, 2020.

Shiraiwa, M. and Seinfeld, J. H.: Equilibration timescale of atmospheric secondary organic

aerosol partitioning, Geophys. Res. Lett., 39, L24801, https://doi.org/10.1029/2012GL054008, 2012.

Simoneit, B. R. T.: Biomass burning - A review of organic tracers for smoke from incomplete combustion, https://doi.org/10.1016/S0883-2927(01)00061-0, 2002.

Xie, M., Hannigan, M. P., and Barsanti, K. C.: Gas/Particle Partitioning of 2-Methyltetrols and Levoglucosan at an Urban Site in Denver, Environ. Sci. Technol., 48, 2835–2842, https://doi.org/10.1021/es405356n, 2014.